

# The impact of electron precipitation on Earth's thermospheric NO production and the drag of LEO satellites

Manuel Scherf[1], Sandro Krauss[2], Grigory Tsurikov[3], Andreas Strasser[2], Valery Shematovich[3], Dmitry Bisikalo[4,3], Helmut Lammer[1], Manuel Güdel[5], and Christian Möstl[6]

[1]Space Research Institute, Austrian Academy of Sciences, Graz, Austria
[2]Institute of Geodesy, Technical University, Graz, Austria
[3]Institute of Astronomy, Russian Academy of Sciences, Moscow, Russian Federation
[4]National Center of Physics and Mathematics, Sarov, Russian Federation
[5]Department of Astrophysics, University of Vienna, Vienna, Austria
[6]Austrian Space Weather Office, GeoSphere Austria, Graz, Austria

**Correspondence:** Manuel Scherf (manuel.scherf@oeaw.ac.at)

**Abstract.** We investigate the response of space weather events on Earth's upper atmosphere over the polar regions by studying their effect on the drag of the CHAMP and GRACE satellites. Increasing solar activity that results in heating and the expansion of the upper atmosphere threatens low Earth orbit (LEO) satellites. Auroral events are closely related to the stellar energy deposition of solar EUV radiation and precipitating energetic electrons, which influence photochemical processes such as the
production of nitric oxide (NO) in the upper atmosphere. To study the production of NO molecules and their influence on the thermospheric structure and satellite drag, we first model Earth's background thermosphere structure with the 1D upper atmosphere model Kompot by considering the incident X-ray, EUV, and IR radiation during selected space weather events. For investigating the effect of electron precipitation in the production of NO molecules in the polar thermosphere, we apply a Monte Carlo model that takes into account the stochastic nature of collisional scattering of auroral electrons in collisions with
the surrounding $N_2$-$O_2$ atmosphere, including the production of suprathermal N atoms. The observed effect of the atmospheric drag on the CHAMP and GRACE spacecraft during the two studied events indicates that a sporadic enhancement of NO molecule production in the polar thermosphere and its IR-cooling capability, which counteracts thermospheric expansion and can lead to an "overcooling" with decreased density after the space weather event, can have a protective effect on LEO satellites. Their production efficiency, however, is highly dependent on the energy flux of the precipitating electrons.

## 1 Introduction

Earth's upper atmosphere interacts with the particles and radiation emitted by the Sun. The absorption of stellar X-ray and extreme ultraviolet (EUV; together abbreviated as XUV) photons heats up, expands, and disperses atmospheric gas into space. In addition to the solar radiation, Earth is occasionally hit by coronal mass ejections (CMEs), i.e., eruptions of plasma and magnetic fields from the Sun's corona. When a CME hits Earth, energy is injected into the magnetospheric system, and
electrons and protons can interact with the atmosphere via the polar regions, through which they can enter the magnetosphere. Such events can cause a geomagnetic storm, during which the CME compresses the magnetosphere on the day side and



the magnetic tail on the night side becomes extended. The charged particles and magnetic fields from the CME and their interaction with the magnetosphere through magnetic reconnection can lead to auroras, disrupt satellite operations, be a danger to manned space missions, and induce electric currents on Earth's surface, which have the capability to disrupt power grids and

radio communications when they collide with the atmospheric atoms and molecules below the exobase (see, e.g., Buzulukova and Tsurutani, 2022, for a review). Additionally, CMEs can also affect the altitude of low Earth orbiting satellites (LEOs) by delivering substantial amounts of additional energy and momentum into the Earth's upper atmosphere, primarily through geomagnetic coupling processes. This significant energy input leads to increased heating of the thermosphere, causing an expansion of these atmospheric layers. Consequently, the enhanced atmospheric density at satellite altitudes results in increased

drag, leading to a measurable storm-induced orbital decay of LEO satellites.

In addition to the above-mentioned space weather effects, the CME-related precipitating electrons interact with molecules in the thermosphere in the cusp regions. These energetic electrons enhance Joule heating (e.g., Zhang et al., 2012) and modify the photochemistry, which results in an enhancement of nitric oxide (NO) production (e.g., Gérard et al., 1991; Shematovich et al., 1991; Barth et al., 1999b, 2003b; Mlynczak et al., 2012; Shematovich et al., 2023), and the production of suprathermal oxygen

atoms (Shematovich et al., 2011). Since NO molecules are IR-coolers in the thermosphere, their increase can lead to an increase in cooling, which acts against the above-mentioned thermospheric heating and expansion of the upper atmosphere. It was found that the thermospheric NO concentration correlates strongly with space weather events and solar activity. The increased production of NO can even lead to an overcooling of the upper atmosphere (Mlynczak et al., 2018; Zhang et al., 2019, 2022; Ranjan et al., 2024) after an initial increase in heating and expansion of the thermosphere (Zhang et al., 2012, 2022). This

complex interplay between heating and cooling presents a significant challenge for accurate thermospheric density prediction during CME events, as underestimating this cooling effect can lead to overestimations of thermospheric expansion and thus larger, less precise forecasts of satellite orbital decay (Krauss et al., 2024). Therefore, a comprehensive understanding and accurate modeling of NO density and its radiative cooling effect are crucial for achieving feasible and reliable forecasts of satellite orbital decay during space weather events. This study, therefore, aims to investigate space weather events and their

responses, such as atmospheric heating and cooling, NO production, and the related impact of the upper atmosphere on the orbit trajectories of satellites.

Section 2 describes the selection and available data of the chosen space weather events. In Section 3, we apply a 1D thermosphere model to the solar activity conditions of the events and model the background atmosphere. We then implement the obtained atmospheric density and temperature profiles into a numerical kinetic Monte Carlo model in Section 4 to study the

NO production by precipitating electrons. After discussing our results in Section 5, we conclude our study in Section 6.

## 2 Selected Coronal Mass Ejection (CME) Events for Analysis

To investigate the response of selected space weather events of Earth's thermosphere at different altitudes, we estimated neutral mass densities by using accelerometer measurements from the Challenging Minisatellite Payload (CHAMP; Reigber et al., 2002) and Gravity Recovery and Climate Experiment (GRACE; Tapley et al., 2004) space missions. Both satellite missions



were dedicated gravity-field missions with on-board accelerometers that measure the impact of non-gravitational forces on the satellite. While both satellite missions maintained near-polar, near-circular orbits, their main difference lay in the varying altitudes they operated during their respective mission lifetimes. Figure 1 shows the corresponding altitudes over the specific mission duration.

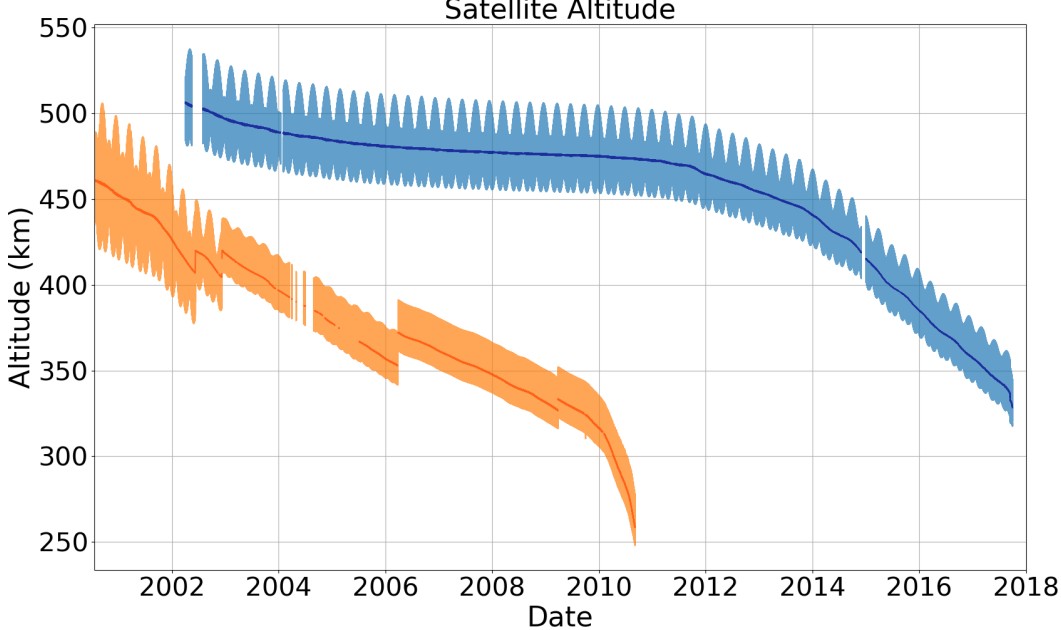

**Figure 1.** Evolution of the altitude of the satellite for the CHAMP (orange) and GRACE (blue) missions over their entire mission duration. Thick lines correspond to the mean altitude per revolution.

For the selected CME events, the altitude of the CHAMP and GRACE spacecraft was on average 360–370 km and 480–
490 km, respectively. For the underlying study, we selected two different periods and investigated neutral mass densities estimated from accelerometer measurements following Krauss et al. (2020), and also derived the storm-induced orbit decay based on these observations (Krauss et al., 2024). The latter can be expressed by the temporal change of the semi-major orbit axis, $a$, and is given by:

$$\Delta a = -\frac{C_{a,x} A_{\mathrm{ref}}}{m} \sqrt{GM\bar{a}} \cdot \psi(e) \int (\rho - \rho_{\mathrm{b}}) dt \qquad (1)$$

In this formulation, $C_{a,x}$ is the ballistic drag coefficient in in-flight direction $x$, $A_{\mathrm{ref}}$ is the area of the satellite, $m$ is the satel-
lite mass, $\rho$ specifies the observed density value, and $\rho_b$ represents the background density of the Earth's thermosphere (Krauss et al., 2018), which we specified as the mean density for two consecutive satellite revolutions prior to the CME arrival time. This specific time is taken from the CME catalog (R&C catalog) provided by Richardson and Cane (2013)[1]. The formulation

---

[1]See https://izw1.caltech.edu/ACE/ASC/DATA/level3/icmetable2.htm.




also includes the Earth's gravitational parameter, $GM$, the mean semi-major axis, $\bar{a}$, averaged throughout the CME event, and an eccentricity function, $\psi(e)$. The latter is nearly 1 for the two satellites under investigation.

## 2.1 Event 1: 9 November 2004

We have chosen this specific CME event (e.g., Trichtchenko et al., 2007) because the thermospheric densities recorded after the perturbations triggered by the CME were significantly lower than before the event. This behavior was visible in the observations of both satellite missions at their respective altitudes. Regarding the position of the spacecraft in space, the solar beta angle, which describes the angle between the orbital plane and the Earth-Sun line, and the median of local solar time (LST) during the event, were as follows: for CHAMP, $\beta = 38°$, LST = 02:35, and for GRACE, $\beta = 0°$, LST = 11:14. Figure 2 shows the impact on the neutral mass densities along the respective satellite trajectories as well as the triggered storm induced orbit decay for CHAMP (left panel) and GRACE (right panel).

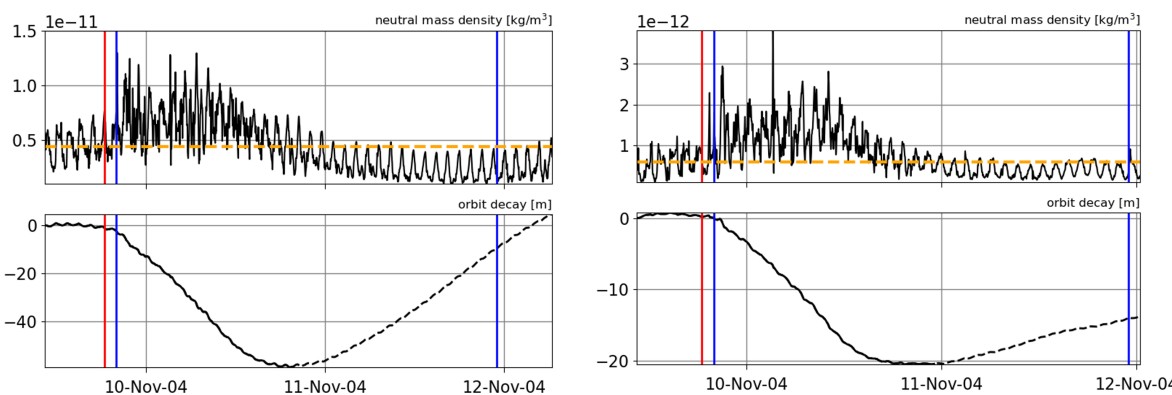

**Figure 2.** Impact of a CME in November 2004 on the CHAMP (left) and GRACE-A satellites (right) at altitudes of 370 km and 490 km, repectively. The top and bottom panels show the neutral density [kg/m$^3$] evolution and the storm-induced orbit decay [m], respectively. Additionally illustrated are the calculated background density (orange line), the start time of the disturbance (red line), i.e., the time of the arrival of a shock or the leading edge of the CME at the Earth, and the observed start and end times of the responsible near-Earth interplanetary CME plasma/magnetic field (blue lines), as specified by the R&C catalog.

The (theoretically) increasing orbital altitude (dashed lines) after the event is caused by the significantly reduced densities compared to the pre-event level ($\rho_b$) and is already an indication that an excessive over-cooling occurred during this time. Mathematically, this implies that the integral in Equation (1) flips the sign in the determination. Although this is not an actual increase in the satellite altitude, the orbit decay parameter can still be used to compare densities before and after the event. One can see from Figure 2 that CHAMP has dropped by about 60 meters due to the increase in density, while GRACE's orbit was affected by about 20 meters. This indicates that CHAMP, with its orbit within the thermosphere, was more strongly



affected by the space weather event because the total atmospheric density near the exobase level, where GRACE was located,
at approximately 500 km altitude, is less dense.

As we know from previous studies (e.g., Chen et al., 2014; Krauss et al., 2018; Oliveira and Zesta, 2019), the interplanetary
magnetic field, $B_z$, plays an important role when analyzing the impact of CME events. However, investigating the observa-
tions from the ACE satellite during that time has not revealed any significant deviations. For this specific event, the SODA
database (Krauss et al., 2023) specifies a value of $B_z$=-31.3 nT, which is completely in line with the expected impact.

Another type of satellite observations we used for the analysis are continuous measurements from the *Thermosphere Iono-
sphere Mesosphere Energetics and Dynamics* (TIMED; see Kusnierkiewicz, 2003, for an overview) satellite's *Sounding of the
Atmosphere using Broadband Emission Radiometry* (SABER) instrument. This instrument measures infrared radiance, which
can be attributed to, e.g., NO in the lower thermosphere. Consequently, this allows the computation of global cooling rates and
radiative fluxes for NO (e.g., Mlynczak et al., 2003). Figure 3 shows geo-reference illustrations of the NO fluxes observed from
the TIMED/SABER satellite above the north and south pole regions. In either case, the measurements represent the complete
altitude-integrated energy flux at a satellite altitude of approximately 625 km. It should be noted that, due to the spacecraft's po-
sition relative to the Sun, SABER will always view the anti-Sun side of the spacecraft, resulting in asymmetric global coverage
over any 60-day period (Russell et al., 1999), leading to a certain amount of visible polar gaps.

A significant increase in the NO concentration on 10 November is visible, particularly around the South Pole. This could
initiate cooling effects, which might explain the lower thermospheric density after the CME event (cf. 2). These in situ
observations provide an excellent basis for further analysis in this study.

## 2.2 Event 2: 15 May 2005

The second investigated CME event occurred on 13 May 2005, through an explosion near sunspot 759. With a transit velocity
of 1270 km/s, the first near-Earth disturbances were recorded on 15 May 2005, at 02:38 UT (Richardson and Cane, 2013,
R&C). Following, e.g., Bisi et al. (2010), this was a rather complex event where we additionally detected a divergent behavior
of the thermospheric densities recorded by the satellites CHAMP and GRACE, which we visualized in Fig. 4.

In contrast to the first event, we do not observe an increase in the satellites' altitude in the storm-induced orbit decay. This
suggests that the density after the event was equal to or greater than before the event. Accordingly, it can be assumed that there
were no increased cooling effects. Additionally, we found that the densities for the two satellites behave differently, especially
during the event, which lasted nearly four days following the specifications in the R&C catalog. During the event, densities
observed at the lower altitude level of CHAMP ($\approx$370 km) show a much slower decrease and, as a result, a significantly longer
lasting storm-induced orbit decay than compared with GRACE at an altitude of about 480 km (cf. 2). In terms of values, an
altitude loss in the order of 112 m and 14 m was observed for the CHAMP and GRACE satellites, respectively. Apart from
the longer lasting decay for CHAMP, the main difference is the actual difference in altitude of about 100 km, which leads to
absolute density differences of a power of ten during that time. Regarding the positions in space, the two satellites are very
similarly located. for CHAMP, $\beta = 38°$, LST = 09:20, and for GRACE, $\beta = 29°$, LST = 09:51.



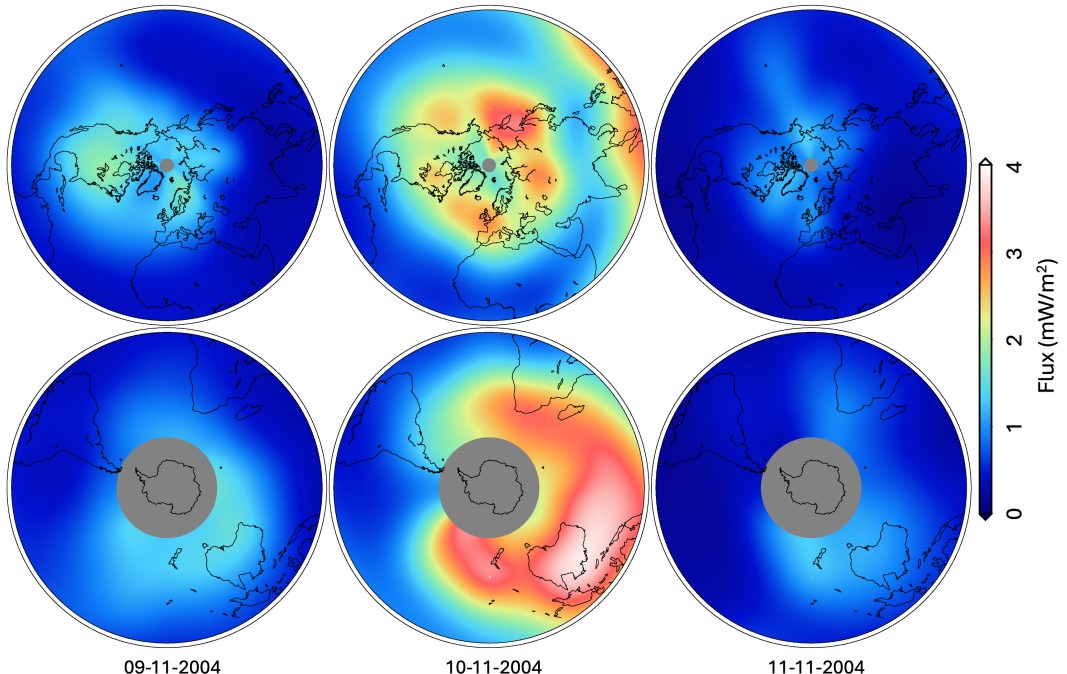

**Figure 3.** Altitude-integrated Nitric oxide (NO) flux observed by the SABER instrument on board the TIMED satellite during the CME event in November 2004. Top and bottom rows illustrate the measurements taken on the northern and southern hemisphere, respectively. These maps are generated by considering all SABER measurements taken during the day, and by then using an interpolation scheme (Mlynczak et al., 2003).

Concerning the measurements by TIMED/SABER during the event, which are illustrated in Figure 5, we see only slightly and selectively increased NO flux values. This agrees well with the assumption that no overcooling took place based on the derived thermospheric density (via Equation 1) in the orbits of CHAMP and GRACE. Crucially, event 2 neither shows a

significant increase in NO production nor any overcooling, whereas event 1 shows both.

## 3   Thermosphere Simulations

### 3.1   Model and input description

To simulate the background thermosphere of Earth for the two events, we apply the 1D upper atmosphere model Kompot (Johnstone et al., 2018), which calculates the thermal and chemical structure of the thermosphere based on the radius and mass

of the planet, the incident solar XUV and IR flux, the homopause temperature, and atmospheric composition. The model was benchmarked with the thermospheres of present-day Earth and Venus and with the atmospheric profiles of the empirical *US Naval Research Laboratory Mass Spectrometer and Incoherent Scatter radar Exosphere* (NRLMSIS) model (Picone et al., 2002). It was also used to simulate Earth's atmosphere in the geologic past (Kislyakova et al., 2020; Johnstone et al., 2021),





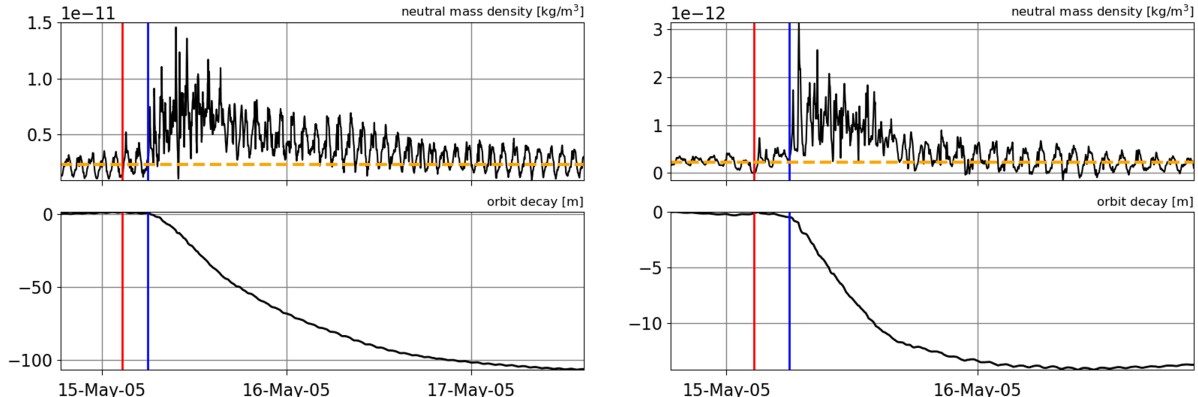

**Figure 4.** Impact of a CME in May 2005 on the CHAMP satellite (left) and GRACE-A satellite (right) 360 km and 480 km. The top and bottom panels show the neutral density [kg/m$^3$] evolution and the storm-induced orbit decay [m], respectively. Additionally illustrated are the calculated background density (orange line), the start time of the disturbance (red line), i.e., the time of the arrival of a shock or the leading edge of the CME at the Earth, and the observed start time of the responsible near-Earth interplanetary CME plasma/magnetic field (blue lines), as specified by the R&C catalog.

and exoplanetary atmospheres of terrestrial planets around highly active stars (Johnstone et al., 2019; Johnstone, 2020; Van
Looveren et al., 2024), illustrating that it can be utilized for a broad range of atmospheric compositions and incident XUV and
infrared fluxes.

Kompot utilizes a network of more than 500 chemical reactions, including more than 50 photoreactions. It takes into account
thermospheric heating from solar XUV (between 1 and 400 nm) and IR (between 1 and 20 $\mu$m) irradiation, from exothermic
chemical reactions, and Joule heating (Johnstone et al., 2018). The model also includes thermal conduction, cooling via in-
frared emission, and energy exchange between neutrals, ions, and electrons, which allows the treatment of neutral, ion, and
electron temperatures separately. It also considers electron heating from collisions with non-thermal photoelectrons produced
by photoionization in the upper atmosphere by assuming that the photoelectrons lose their energy locally where they are created
(Johnstone et al., 2018). However, in addition to this local treatment of internally produced photoelectrons, the precipitation of
externally produced electrons from the Earth's polar regions onto the upper atmosphere is not included in Kompot, although
they have already been shown to modulate the thermospheric structure by either enhancing Joule heating (e.g., Zhang et al.,
2012) or fueling NO production, which can lead to an increase in IR cooling by NO (e.g., Mlynczak et al., 2018). This implies
that the model can generally account for the Sun's irradiation and changes therein, but not for the Sun's or the Earth's magnetic
field and plasma environment.

As input for our event simulations, we take the atmospheric composition and neutral temperature at the homopause (assumed
to be at 80 km, i.e., the lower spatial boundary of our model) from the NRLMSIS empirical model[2] (Picone et al., 2002). The

---

[2]The model can be run online; see https://ccmc.gsfc.nasa.gov/models/NRLMSIS~00/.



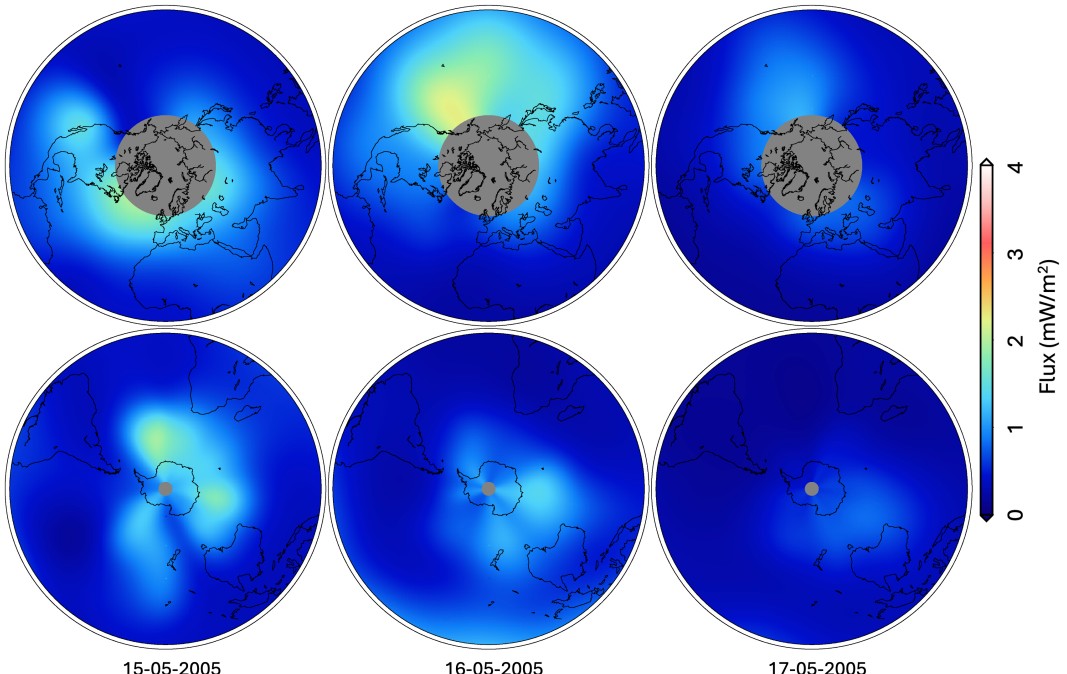

| | | |
|---|---|---|
| 15-05-2005 | 16-05-2005 | 17-05-2005 |

**Figure 5.** Nitric oxide (NO) flux observed by the SABER instrument on board the TIMED satellite during the CME event in May 2005. The top and bottom rows illustrate the measurements taken on the northern and southern hemispheres, respectively. These maps are generated by considering all SABER measurements taken during the day, and by then using an interpolation scheme (Mlynczak et al., 2003).

NRLMSIS model is an empirical model of Earth's atmosphere that extends from the surface to the exobase level and describes the average observed temperature behavior, the densities of the eight main species, and the mass density via a parametric analytic formulation.

We take the values at a zenith angle of $\Theta = 66°$ at 12:00 UTC, since this angle gives the best representation of the atmosphere of the present-day Earth (see Johnstone et al., 2018, for details on benchmarking of Kompot). To obtain the XUV flux of the respective event days, we take daily averaged observational and model data by the SEE instrument of the TIMED spacecraft (Woods et al., 2005). For the IR spectrum between 1 and $20\,\mu$m, a simple black-body spectrum with a temperature of 5777 K is assumed (Johnstone et al., 2018). The exact shape and intensity of the IR spectrum, however, do not matter, as its influence on the Earth's upper atmosphere, in contrast to Venus and Mars, is negligible due to both the low total atmospheric density

and mixing ratio of $CO_2$ in thermosphere, because of which the IR irradiation is mostly absorbed in lower atmosphere below our lower model boundary of 80 km. The main driver of differences in thermospheric structure and density, as simulated with Kompot, will hence be the incident XUV flux.

   Table 1 lists the input parameters for our Kompot runs, i.e., the X-ray surface flux, $F_X$ as derived from the NASA Thermosphere Ionosphere Mesosphere Energetics and Dynamics (TIMED) and Solar EUV Experiment (SEE) databases[3], the ho-

---

[3]Link for the TIMED-SEE informations and databases: https://lasp.colorado.edu/see/.





**Table 1.** Input parameters from TIMMED/SEE databases and NRLMSIS model (see main text) into Kompot.

|  | Event 1: 2004-11-09 | Event 2: 2005-05-15 |
| --- | --- | --- |
| $F_{\mathrm{X}}$ ($10^{-4}$ W/cm$^2$) | 4.88 | 4.16 |
| $T_{\mathrm{hp}}$ (K) | 189.9 | 187.8 |
| $n_{\mathrm{hp}}$ ($10^{14}$ cm$^{-2}$) | 4.16 | 2.31 |
| N$_2$ ($n_{\mathrm{N_2}}/n_{\mathrm{hp}}$) | 0.787 | 0.783 |
| O$_2$ ($n_{\mathrm{O_2}}/n_{\mathrm{hp}}$) | 0.204 | 0.208 |
| CO$_2$ ($10^{-6}\, n_{\mathrm{CO_2}}/n_{\mathrm{hp}}$) | 400 | 400 |
| N ($10^{-10}\, n_{\mathrm{N}}/n_{\mathrm{hp}}$) | 2.23 | 1.33 |
| O ($10^{-5}\, n_{\mathrm{O}}/n_{\mathrm{hp}}$) | 1.08 | 0.45 |
| H ($10^{-8}\, n_{\mathrm{H}}/n_{\mathrm{hp}}$) | 6.68 | 6.69 |
| He ($10^{-6}\, n_{\mathrm{He}}/n_{\mathrm{hp}}$) | 5.46 | 5.47 |
| H$_2$O ($10^{-6}\, n_{\mathrm{H_2O}}/n_{\mathrm{hp}}$) | 6.0 | 6.0 |
| Ar ($10^{-3}\, n_{\mathrm{Ar}}/n_{\mathrm{hp}}$) | 9.34 | 9.31 |

mopause temperature and density, $T_{\mathrm{hp}}$ and $n_{\mathrm{hp}}$, as well as the homopause mixing ratios from various neutral species. All mixing ratios are derived from NRLMSIS except for CO$_2$ and H$_2$O, which were always assumed to be 400 and 6 ppm at the homopause, respectively.

As a next step, we implement the above-described parameters (see also Table 1) and model the corresponding upper atmosphere structures for the two selected events. The potential additional effect of electron precipitation is then separately discussed in Section 4, for which the atmospheric profiles simulated with Kompot serve as the background atmosphere.

### 3.2 Thermosphere structure based on the Kompot runs

Figure 6 shows the atmospheric densities of selected neutrals (left panel) and ions (right panel) in the upper atmosphere of the Earth for the two events as simulated with Kompot (based on the daily averaged XUV flux and homopause densities as taken from NRLMSIS; see Table 1). Event 2 shows an exobase altitude of about 435 km with an exobase density of $\sim 7.5 \times 10^7$ cm$^{-3}$. Event 1, on the other hand, has a significantly higher exobase altitude of around 500 km and hence higher densities in the upper atmosphere. For the orbits of CHAMP and GRACE at 380 and 490 km, these are $\sim 3.2 \times 10^8$ cm$^{-3}$ and $\sim 6.7 \times 10^7$ cm$^{-3}$, respectively. For event 2, we only get the orbital densities for CHAMP at an orbit around 370 km of $\sim 2.1 \times 10^8$ cm$^{-3}$ as the exobase altitude is below the GRACE orbit. Interestingly, event 1 shows lower NO densities (solid thick green line) than event 2 between 85 and 110 km of the events, although it is higher at all other altitudes.

Based on the input data, it is not surprising that event 1 shows higher upper atmosphere densities and a larger expansion. For this event, the X-ray surface flux, $F_{\mathrm{X}}$, and hence also the entire XUV flux, as well as the base densities at the lower model boundary of 80 km are all larger than for event 2, with its base density being almost twice as high.



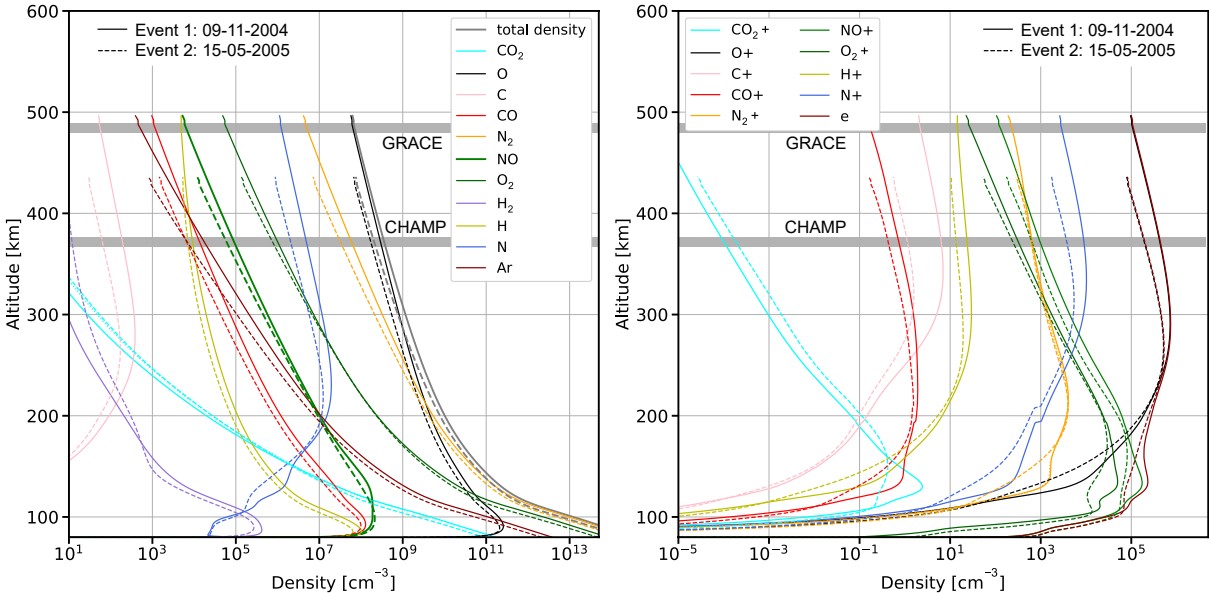

**Figure 6.** Neutral (left) and ion (right) densities of selected species for the two events as simulated with Kompot. The horizontal grey areas illustrate the orbits of CHAMP and GRACE (i.e., about 380 and 490 km for event 1 and 370 and 480 km for event 2, respectively).

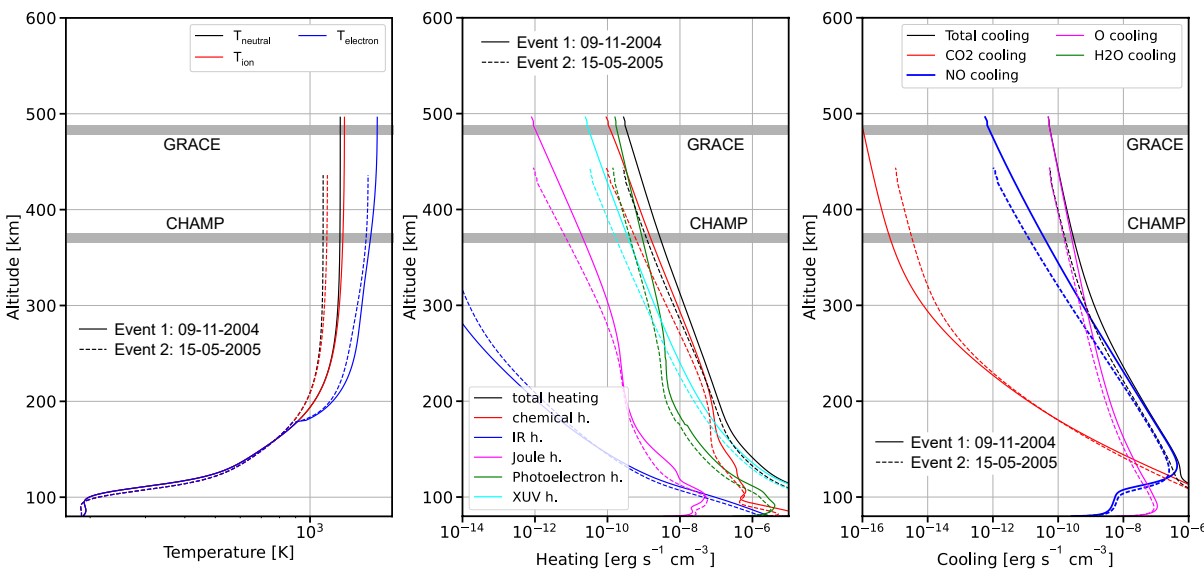

**Figure 7.** Temperature profiles (left), heating (middle), and cooling (right panel) processes for the two events as simulated with Kompot. The thicker blue lines in the right panel illustrate cooling via NO. The horizontal grey areas again illustrate the orbits of CHAMP and GRACE (i.e., about 380 and 490 km for event 1 and 370 and 480 km for event 2).





As further expected, event 1 also has the hotter temperature in the upper atmosphere, as can be seen in the left panel of Fig 7, which shows the neutral, ion, and electron temperatures of the two events. The neutral temperature of event 1 reaches above 1.200 K, whereas it is around 1.100 K for event 2. The middle panel of this figure highlights the various heating sources, showing that photoelectron and chemical heating are the dominating heating sources in the orbits of CHAMP and GRACE, followed by XUV heating. IR heating in the upper atmosphere is almost negligible, as expected (see above).

The right panel, finally, shows the cooling processes in the thermosphere. Here, we highlight that NO cooling already dominates the lower part of the thermosphere between about 120 and 300 km in both events, even though these runs do not consider NO production via electron precipitation (see next section). The upper part, on the other hand, is dominated by O cooling, since electron precipitation mostly affects the lower layers of the thermosphere. Cooling via $CO_2$ dominates below $\sim$120 km but is insignificant in the rest of the thermosphere due to the very low mixing ratio (assumed to be 400 ppm at the lower boundary), which strongly decreases with altitude.

Next, we evaluate how electron precipitation affects the NO production in the upper atmosphere, and hence atmospheric (over)cooling for the two events. For this, we take the background atmosphere as simulated with Kompot and feed it into the Monte Carlo model described in the next section.

## 4 The formation of NO molecules in the Earth's thermosphere

The precipitation of energetic 1-10 keV electrons of magnetospheric origin into the polar regions of the Earth's thermosphere can lead to its heating, changes in the chemical composition, and the formation of suprathermal particles with kinetic energies more than an order of magnitude higher than the thermal energy of the surrounding gas. With the increase of geomagnetic activity, the additional heating of the atmosphere and the formation of suprathermal oxygen atoms induced by electron precipitation can affect the drag of LEO satellites (Wilson et al., 2006; Shematovich et al., 2011; Krauss et al., 2012).

It is known from observations of the Earth's atmosphere (Barth et al., 2003a) that electron precipitation is the main source of NO molecule production in the polar regions. This molecule is an important minor atmospheric constituent in the lower thermosphere because of its radiative and chemical properties (e.g., Barth et al., 1999a). Satellite measurements have shown that the NO production with its maximum near 110 km (e.g., Barth, 1992) and its variability correlate well with solar activity and space weather events. Collisions of auroral electrons with molecular nitrogen lead to its dissociation, ionization, and dissociative ionization:

Reaction (1):     $N_2 + e^- \rightarrow N(^4S) + N(^2D),$     (2)

Reaction (2):     $N_2 + e^- \rightarrow N_2^+ + 2e^-,$     (3)

Reaction (3):   $N_2 + e^- \rightarrow N^+ + N(^4S, ^2D) + e^-,$     (4)

These processes are the main drivers of odd nitrogen chemistry (the system of chemical kinetic reactions in which NO is produced and lost) in the Earth's polar thermosphere. The interaction of the dissociation products, nitrogen atoms in the ground and metastable states, $N(^4S)$ and $N(^2D)$ ($\Delta E = 2.38$ eV), with molecular oxygen $O_2$, is the main source of NO





formation (Gerard and Barth, 1977), (Barth, 1992), (Barth et al., 2003a):

Reaction (4):     $N(^2D) + O_2 \rightarrow NO + O,$                                      (5)

Reaction (5):     $N(^4S) + O_2 \rightarrow NO + O,$                                      (6)

Reaction (4) has no activation energy, while Reaction (5) is characterized by an energy barrier of 0.3 eV. Its rate, therefore, strongly depends on temperature.

Under quiet geomagnetic conditions, the main source of NO formation in the equatorial and mid-latitude regions of the Earth's thermosphere is photoelectrons formed due to the absorption of soft X-ray radiation from the Sun by the atmospheric gas (Barth, 1992; Barth et al., 1999a, 2003a). However, with increasing geomagnetic activity, NO can be transported from polar latitudes to mid- and equatorial latitudes, presumably due to the meridional wind (Barth et al., 2003a; Dothe et al., 2002; Sætre et al., 2007). The NO molecule, therefore, in addition to being an effective cooler of the atmosphere, is an indicator of

solar and geomagnetic activity (Barth et al., 2004; Mlynczak et al., 2015), as well as an indicator of horizontal mass transfer in the upper atmosphere (Barth et al., 2003a). Additional sources of NO are:

     a) Joule heating, which affects the temperature-sensitive Reaction (5) and can lead to vertical transport of NO (Siskind et al., 1989b, a);

     b) suprathermal nitrogen atoms $N_{hot}(^4S)$.

Suprathermal nitrogen atoms are formed in Reaction (1) with an excess kinetic energy (Cosby, 1993), and the interaction of $N_{hot}(^4S)$ with $O_2$ is an efficient non-thermal channel for NO production, because the suprathermal nitrogen atoms can overcome the energy barrier of this reaction (Shematovich et al., 1991, 2023, 2024; Gérard et al., 1991, 1995, 1997):

Reaction (6): $N_{hot}(^4S) + O_2 \rightarrow NO + O,$                                         (7)

### 4.1 Calculation of NO production for the two selected events

Since the NO molecule is an effective IR-cooler in the thermosphere, its production during electron precipitation can compensate for the heating and expansion of the upper atmosphere, leading to an overcooling after the event and reducing the drag force acting on the LEO satellites with polar orbits. Therefore, in this Section, our attention is focused on determining the contribution of the electron precipitation process to the production of NO molecules in the Earth's thermosphere for the geomagnetic storms that are related to the two space weather events considered earlier: event 1 from 9 November 2004 (Ap = 140)

and event 2 15 May 2005 (Ap = 87).

     For this purpose, the odd nitrogen chemistry model presented in Shematovich et al. (2024) is used. The model is based on the equations of chemical kinetics, molecular diffusion, and eddy diffusion. The numerical solution to the problem is found using the method of splitting by physical processes. Therefore, the following equations are solved step by step:

     a) The system of chemical kinetics equations: 19 reactions with components NO, $O_2$, O, $N(^4S, ^2D)$, $NO^+$, $N_2^+$, $O_2^+$, $O^+$,

240          $N^+$, and $e^-$ (see Table 1 in Shematovich et al., 2024), which describe the odd nitrogen chemistry in the atmosphere.



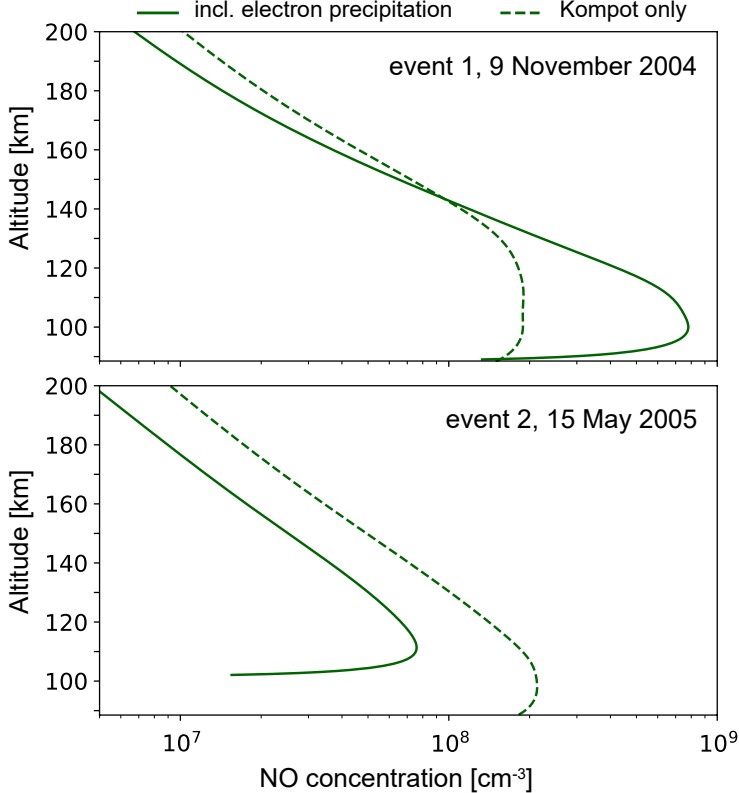

**Figure 8.** Altitude distributions of NO molecule concentrations in Earth's thermosphere for the two geomagnetic storms: event 1, 09 November 2004 (upper panel), and event 2, 15 May 2005 (lower panel). Solid lines correspond to NO concentrations as calculated with the model of Shematovich et al. (2024), where the electron precipitation is included. The dashed lines correspond to the results obtained with Kompot, which only includes the daily average of the solar XUV flux as an input.

The solution is found using the open-source software package CVODE (Cohen et al., 1996) for systems of ordinary differential equations;

b) The diffusion equation for NO and $N(^4S)$. The contribution of diffusion to the NO altitude distribution is comparable to the contribution made by the NO chemistry, since both the lifetimes of an NO molecule against chemical destruction and diffusive transport are around one day (Bailey et al., 2002). The solution is found using the Crank-Nicolson method. Details of the numerical implementation are described in Johnstone et al. (2018).

The calculations are carried out until a steady state is established. The input data in the model are:

a) the altitude distributions of the concentrations of neutrals, ions, electrons, and their temperatures for the background atmosphere are calculated using Kompot (see previous Section);



b) the rates of dissociation, ionization, and dissociative ionization of molecular nitrogen by electron impact (Reactions 1-3).
To calculate these rates, we used the kinetic Monte Carlo model for precipitating electrons presented in Bisikalo et al.
(2022). Based on the solution of the Boltzmann equation by the kinetic Monte Carlo method, this model describes the
evolution of precipitating energetic electrons as a result of elastic, inelastic, and ionization collisions with the surrounding
atmospheric gas (see Bisikalo et al., 2022, Appendix A). The atmospheric gas is assumed to be characterized by a local

Maxwellian velocity distribution. The main outputs of this model are the electron energy spectra in each atmospheric
layer and the altitude distribution of the integral downward and upward electron energy fluxes. Knowing these quantities,
as well as the $N_2$ concentration in the atmosphere and the cross sections of Reactions (1-3) (Tabata et al., 2006; Itikawa,
2006; Jackman et al., 1977), we can calculate the rates of these processes.

Our calculations focused on determining how, on average, the electron precipitation affects the NO molecule concentration

in the thermosphere. Therefore, at the upper boundary of the computational domain (700 km), we specified the distribution of
precipitating electrons by the kinetic energy spectrum using a Maxwellian function with the characteristic energy $E_0 = E_{\mathrm{m}}/2$
($E_{\mathrm{m}}$ is the mean electron kinetic energy) and assumed an isotropic pitch-angle distribution for precipitating electrons relative
to the geomagnetic field lines. By assuming a Maxwellian distribution, we follow the recommendation of Hardy's empirical
model of auroral electron precipitation (see, Hardy et al., 1985). The ratio between $E_0$ and $E_{\mathrm{m}}$ further follows directly from

the used Maxwellian energy spectrum.

At the upper boundary, the energy flux of precipitating electrons, $Q_0$, was also specified. To determine $E_0$ and $Q_0$ at the
upper boundary, we used NOAA DMSP satellite measurements for the precipitating electrons (Redmon et al., 2017). For this,
we averaged over the northern polar oval the $E_0$ and $Q_0$ values that were measured by DMSP-F13, F14, F15, and F16 satellites[4]
during the considered events. As a result of averaging, we obtained the following $E_0$ and $Q_0$ values:

– event 1, 9 November 2004 – for DMSP-F15: $E_0 = 1.279$ keV and $Q_0 = 1.0$ $\mathrm{erg\,cm^{-2}\,s^{-1}}$;

– event 2, 15 May 2005 – for DMSP-F13: $E_0 = 0.273$ keV and $Q_0 = 0.3$ $\mathrm{erg\,cm^{-2}\,s^{-1}}$

Figure 8 shows the altitude profiles of the NO concentration from our calculations for the two geomagnetic storms under
study. The NO concentration profiles calculated using Kompot are also shown for comparison. We reiterate that Kompot only
considers the XUV flux as an input and therefore mostly underestimates nitric oxide formation. Our results can hence be

analyzed as follows:

– **event 1, 09 November 2004** – This event is characterized by the highest $E_0$ value among the presented cases. It is
known (Bailey et al., 2002; Shematovich et al., 2024) that the maximum NO concentration is reached in the region of
the peak energy deposition of precipitating electrons, where the rates of the Reactions (1-3) reach their peak values.
The higher the mean kinetic energy of electrons, the deeper they penetrate the atmosphere. Therefore, among the two

considered events, the maximum NO concentration for the storm on 09 November 2004 is located at the lowest altitude,
100 km (see the upper panel of Figure 2). The peak NO concentration of $7.8 \times 10^8$ $\mathrm{cm^{-3}}$, as obtained by using the

---

[4]http://sd-www.jhuapl.edu/Aurora; https://registry.opendata.aws/dmspssj; https://dmsp.bc.edu



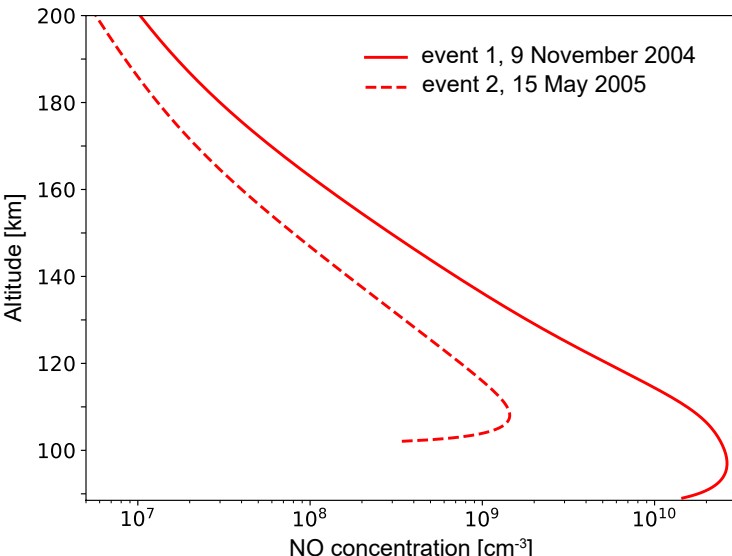

**Figure 9.** Altitude distributions of NO molecule concentrations calculated with the addition of the non-thermal NO-production channel shown in Reaction (6) for the two geomagnetic storms: 2004-11-09 (solid lines) and (dashed lines).

model of Shematovich et al. (2024), is 4 times larger than the concentration obtained from Kompot. This difference shows the importance of considering electron precipitation when modeling the NO concentration. It is known that the rate of atmospheric cooling due to IR radiation of NO at a wavelength of $5.3$ microns is proportional to the concentration

of this molecule (Oberheide et al., 2013). Therefore, such a difference in the NO concentration may lead to different estimates of the heat balance in the thermosphere. The high concentration of NO due to the strong geomagnetic storm on 09 November 2004 could potentially lead to a compensation for the heating and expansion of the atmosphere during the event and a subsequent overcooling occurring after the event. Both may, consequently, lead to a decrease in the drag force acting on the satellites in question. As described in Section 2.1 and as can be seen in Figure 1, a strong decrease in

atmospheric density at the orbits of both CHAMP and GRACE was detected directly after the event took place, thereby indicating an excessive overcooling. This agrees with the strong NO production via electron precipitation as found by our model. We note, however, that our model only provides a static snapshot but no time-resolved evolution, which is beyond the scope of the present study.

– **event 2, 15 May 2005** – This event is characterized by much lower values for both $E_0$ and $Q_0$ compared to event 1. Due

to the small value of $E_0 = 0.273\,\mathrm{keV}$, the maximum concentration of NO is located quite high, at $111\,\mathrm{km}$. It is known that the dependence of the maximum value of the NO concentration on the energy flux of precipitating electrons is almost linear in the region of small $Q_0$ (Bailey et al., 2002; Shematovich et al., 2024; Tsurikov et al., 2024). That is why the peak value of the concentration of this molecule, $7.5 \times 10^7\,\mathrm{cm}^{-3}$, has a lower value compared to event 1. It is even 2 times less than the peak value obtained with Kompot. The NO concentration in the thermosphere could therefore be



insufficient to compensate for any heating and expansion of the atmosphere, or even to induce a subsequent overcooling, during the geomagnetic storm on 2005-05-15. This agrees with TIMED/SABER observations of the NO flux, which was observed to be insignificant (see Figure 5). It also agrees with the thermospheric density evolution of this event, which shows no indication of any overcooling after the event (see Figure 4).

The mentioned "compensation" effect (see above) can be even stronger when implementing the non-thermal channel of

NO formation (Reaction 6). Figure 9 shows the NO concentration profiles calculated for the events under consideration with additional consideration of Reaction (6). Compared to Figure 8, it is clear that the peak concentration of this molecule can increase by almost 2 orders of magnitude. The efficiency of the non-thermal channel, however, under conditions of electron precipitation with small $Q_0 \sim 1.0$ $\mathrm{erg\,cm^{-2}\,s^{-1}}$ in the Earth's atmosphere can be significantly lower (Shematovich et al., 2024). For the correct calculation of non-thermal nitric oxide formation in the Earth's thermosphere, it is necessary to carry

out non-stationary calculations, which will be done in our further work.

## 5   Discussion

Our results show that the concentration of the NO molecule in the polar and middle latitudes of the Earth's thermosphere is largely determined by the precipitation of energetic electrons during disturbed geomagnetic conditions. The higher the mean kinetic energy and energy flux of precipitating electrons, the lower the NO maximum altitude and the higher the absolute value

of the peak NO concentration, respectively. Since NO is an effective coolant in the upper atmosphere, an increase in the NO concentration during geomagnetic storms can compensate for the heating and even lead to the observed "overcooling" effect that we see in event 1 but not in event 2, thereby potentially reducing the drag force acting on the LEO satellites.

Therefore, it is important to correctly model NO formation as a response to increased geomagnetic activity in LEO satellite orbit prediction models. As our modeling results show, simulating the thermosphere purely based on the incident irradiation

from the Sun, or more generally from the host star, may not be sufficient to account for the NO production, and hence for the thermospheric structure during and after CME events, as electron precipitation can induce a significant production of NO in the upper atmosphere if the incident electron flux, $Q_0$, is sufficiently high. Based on the observational data from the DMSP satellites, our model chain for the thermosphere correctly predicts a large production of NO during event 1 but no significant production during event 2, both in agreement with TIMED/SABER observations. This confirms the role of electron

precipitation in the production of thermospheric NO during specific CME events. But it also shows that specific conditions must be met for the NO production to significantly increase.

More specifically, the reason for the difference in the electron precipitation impact on the NO production is related to the strengths of the energy flux of the electrons, which is about 3 times lower for event 2 compared to event 1. As can be seen from our calculations, the electron energy, $E_0$, determines the depth (or the peak height) of the auroral electron penetration, and the

corresponding energy flux, $Q_0$, determines the magnitude of the NO production at the peak of the height profiles. Different energy fluxes of precipitating electrons in auroral regions are mainly related to variations in magnetospheric processes such as reconnection events, plasma instabilities, atmospheric interactions, and variations in the wave-particle interaction processes





such as acceleration and scattering of electrons. We note that these factors influence the energy distribution and direction of electrons and hence their energy fluxes when they travel from the magnetosphere to the thermosphere.

Moreover, it should also be mentioned that the choice of the $E_0$ and $Q_0$ values according to the DMSP satellite data is a somewhat tricky procedure. These satellite data are measured with high separation in time and space by several DMSP satellites. Accordingly, the values of $E_0$ and $Q_0$ are selected by averaging over a large set of measurements during the crossing of the polar oval by the CHAMP and GRACE satellites. Unfortunately, this averaging procedure is not an easy procedure, since, for example, the points of entry and exit from the polar oval are not fully known. In future studies, we plan to investigate

the effect of different $E_0$ and $Q_0$ values on the NO production in more detail, by including also data from commercial space weather programs, such as those described in Redmon et al. (2017).

As mentioned above, our results further indicate that the overcooling effect can be larger if the non-thermal channel that produces suprathermal $N_{hot}(^4S)$ and its related NO formation of Reaction (6) is considered. We note that the calculated non-thermal NO production channel caused by auroral electron precipitation is a sporadic input into the odd nitrogen chemistry

over Earth's polar regions. Moreover, the downward transport of NO molecules by the polar vortex to the mesosphere and stratosphere should be considered in the prognosis of climate change caused by solar forcing.

Finally, it is important to note that studies of non-thermal NO production caused by the auroral electron precipitation enhance our understanding of the role of the suprathermal atom fraction in the odd nitrogen chemistry in $N_2$-$O_2$-dominated atmospheres of terrestrial planets in general. As this chemical pathway can cool the upper atmosphere, it could also be an

important factor to be considered for Earth-like atmospheres that receive a larger XUV irradiation and stellar wind from their host star. Earth-like atmospheres with minor $CO_2$ mixing ratios heat and expand significantly for relatively low XUV fluxes (e.g., Tian et al., 2008a, b; Johnstone et al., 2021; Scherf et al., 2024; Van Looveren et al., 2024), an effect that could have eroded Earth's atmosphere during the Archean eon (Johnstone et al., 2021), and critically affects habitability in general (Scherf et al., 2024; Van Looveren et al., 2025). Odd nitrogen chemistry via electron precipitation, as modeled in this study, could alter

atmospheric stability due to its effect on the thermospheric structure. Future studies, however, are needed to investigate its role in atmospheric erosion and, more generally, in habitability.

## 6    Conclusions

We studied two selected space weather events and atmospheric responses to the orbits of the CHAMP and GRACE satellites against upper atmosphere expansion caused by thermospheric heating. Our investigation indicates that, depending on the energy and the related energy flux of the precipitating electrons over the polar regions, an enhanced production of IR-cooling NO

molecules in the thermosphere occurs. These molecules counteract thermospheric heating by the XUV radiation and Joule heating and can lead to an overcooling of the upper atmosphere after the event. The observed atmospheric drag of the CHAMP and GRACE spacecraft during the two studied events agrees with theoretical findings that point out the importance of NO as an IR-cooler of Earth's upper atmosphere over the cusp regions. Moreover, we show that the electron impact dissociation of

$N_2$ is an important source for the production of suprathermal nitrogen atoms, where $N_{hot}(^4S)$ atoms significantly increase the



non-thermal production of NO molecules in the auroral regions. The related overcooling of the thermosphere caused by these nitric oxide molecules has a protective effect on LEO satellites. Furthermore, similar space weather processes will also occur at other terrestrial-type planets with Earth-like atmospheres (Scherf et al., 2024). Their host stars' XUV flux, stellar wind, and, as shown in this study, the related electron precipitation onto an Earth-like planet's $N_2$-$O_2$-dominated atmosphere may affect

their long-term stability and habitability.

*Code and data availability.*    The codes (Kompot and the Monte Carlo model) are not freely available; however, we encourage contacting the authors for collaborations, i.e., Manuel Güdel (manuel.guedel@univie.ac.at) for Kompot and Valery Shematovich (shematov@inasan.ru) for the Monte Carlo model. The data is available upon request to the authors. For general request, please contact Manuel Scherf (manuel.scherf@oeaw.ac.at.

*Author contributions.*    MS, HL, and SK conceptualized the study. MS prepared the input data for thermospheric simulations, modeled the

thermosphere, and evaluated the model data. GT, VS, and DB modeled electron precipitation. SK and AS selected the two events and prepared the CHAMP, GRACE, and TIMED/SABER data. MS, GT, SK, and HL wrote the initial manuscript. MG and CM supported the preparation of the study.

*Competing interests.*    None of the author's have competing interest.

*Acknowledgements.*    MS, SK, AS, and HL acknowledge support by the Austrian Science Fund (FWF) P33620-N. MS and HL acknowledge

FWF for the support of the VeReDo research project, grant I6857-N.' VS, DB, and GT were supported by the Russian Science Foundation under Grant 22-12-00364-p. This work is supported by ERC grant (HELIO4CAST, 10.3030/101042188). Funded by the European Union. Views and opinions expressed are, however, those of the author(s) only and do not necessarily reflect those of the European Union or the European Research Council Executive Agency. Neither the European Union nor the granting authority can be held responsible for them.



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
