# Peer review of "The impact of electron precipitation on Earth's thermospheric NO production and the drag of LEO satellites"

_EGUsphere, 2025_

## Referee Comment (RC1)

Review report on the manuscript "The impact of electron precipitation on Earth's thermospheric NO production and the drag of LEO satellites", submitted to ANGEO by Scherf et al. for the consideration of publication.

**Manuscript summary**

The authors combine 1D Kompot thermosphere runs (background atmosphere) with a kinetic Monte-Carlo model of precipitating electrons (Shematovich et al. approach) to estimate NO production during two CME-drive storm events and examine consequences for thermospheric cooling and satellite drag. They compare Kompot-only results vs. calculations including electron precipitation and compare with SABER observations and CHAMP/GRACE density-derived orbital decay.

**Overall manuscript recommendation**

This is an interesting and valuable manuscript. The modelling approach and the data comparisons are appropriate, and the results are relevant for satellite drag/space-weather forecasting communities. The main scientific message — that precipitation-driven NO can cause overcooling and can therefore affect thermospheric densities and subsequent satellite orbital decay — is supported by the modelling and SABER/accelerometer evidence. However, I recommend minor–major revisions before acceptance: the authors should (i) explicitly connect the results to empirical models and storm recovery mechanics (see Major point #1), (ii) discuss possible NO cooling timing on model predictions, and (iii) clarify the use of SABER data..

**Major comments**

1. Discussion of possible NO effects on empirical thermospheric neutral mass density models. Although the manuscript clearly shows (and discusses) that externally precipitating electrons fuel NO production and that increased NO can drive infrared cooling and overcooling, a link to empirical models during storm recovery should be made more explicit. The manuscript benchmarks Kompot against the empirical NRLMSIS model and repeatedly notes that Kompot does not include externally precipitating electrons, i.e., Kompot (and many empirical/parametric approaches) therefore will miss NO produced by precipitating electrons. This important limitation is explicitly stated. However, the paper does not yet clearly walk the reader through the specific mechanism and timing by which omission of precipitation (and the resulting NO) leads to errors in empirical thermospheric models during the recovery phase (when NO cooling can cause densities to fall below pre-storm levels). The paper mentions that underestimating cooling can overestimate expansion/drag (thus implying impacts on forecasting), but an explicit paragraph that: (a) names typical empirical models (NRLMSIS, etc.), (b) explains how those models are forced/parametrized during storms and recovery, and (c) quantifies (or gives literature evidence for) the size and timing of the bias during recovery would strengthen the manuscript. See lines where the implication is implied but not spelled out.

In this case, I recommend the authors add a short subsection in Discussion explicitly entitled something like "Implications for empirical models and storm recovery" that explains why omission of precipitation-driven NO leads to errors specifically during the recovery phase

(timing: NO lifetime/diffusion ~1 day is mentioned and important). Also, if possible, provide a short numerical estimate or point to literature values (see below) on how big the cooling bias can be and whether it systematically moves empirical model outputs relative to observations.

There has been previous work done on NO cooling effects on empirical models. For example, Oliveira and Zesta (2019) noted that the lack of NO information in the Jacchia-Bowman 2008 (JB2008) model is most likely a major source for density errors during recovery phase of storms, particularly during extreme events. Licata et al. (2021) also observed the same features with CHAMP and GRACE data, but they noted that the HASDM (High Accuracy Satellite Drag Model) was able to capture cooling effects due to NO (recovery) and $CO_2$ (pre-storm) phases. Oliveira et al. (2021) also noted with a superposed epoch analysis that HASDM was able to capture NO effects and even an overcooling effect supported by observations (CHAMP and GRACE), but JB2008 failed miserably during the recovery phase of the storm. One more. Zesta and Oliveira (2019) were able to quantify the timing of such cooling effects, noting that the thermosphere heats and cools faster for the more extreme geomagnetic storms. I think the NRL-MSIS results showed by the authors are expected, since the lack of NO effects also have profound impacts on model results during storm recoveries in the case of JB2008. I think this discussion should be added to support the authors' conclusion stating that, e.g., "[…] NO molecules have [not has] protective effect on LEO satellites." (line 367)

Licata, R. J., Mehta, P. M., Tobiska, W. K., Bowman, B. R., & Pilinski, M. D. (2021). Qualitative and Quantitative Assessment of the SET HASDM Database. *Space Weather*, 19, e2021SW002798. https://doi.org/10.1029/2021SW002798

Oliveira, D. M., Zesta, E., Mehta, P. M., Licata, R. J., Pilinski, M. D., Kent Tobiska, W., & Hayakawa, H. (2021). The current state and future directions of modeling thermosphere density enhancements during extreme magnetic storms. *Frontiers in Astronomy and Space Sciences*, 8 (764144). https://doi.org/10.3389/fspas.2021.764144

Zesta, E., & Oliveira, D. M. (2019). Thermospheric heating and cooling times during geomagnetic storms, including extreme events. *Geophysical Research Letters*, 46 (22), 12,739-12,746. https://doi.org/10.1029/2019GL085120

2. The Kompot runs are steady /1-D background solutions (daily averaged XUV forcing and homopause boundary from NRLMSIS) and the NO production via the Shematovich model is solved to steady-state. The manuscript acknowledges Kompot does not include precipitation and that the NO/diffusion lifetimes (~1 day) matter. However, I strongly recommend the authors make clearer (in Methods or Discussion) the limits of these steady/1-D assumptions for transient recovery behavior (e.g., how the one-day chemical/diffusion timescale compares to recovery timings). The authors could tie such discussion with the heating and cooling times provided by Zesta and Oliveira (2019). Advise that full 3-D, time-dependent runs would be needed to fully capture spatial and temporal evolution of NO cooling during recovery.

3. SABER NO flux maps are used; the manuscript states that event-1 shows increases consistent with overcooling while event-2 does not. This is good. However, consider adding a brief note on the limits of SABER sampling (anti-sun viewing, gaps, hemispheric coverage) and how that

affects the interpretation of polar NO enhancements vs. global effects — the paper already points this out (good), but a sentence tying that observation limitation into inference about recovery would help.

4. I recommend the authors also cite Knipp et al. (2017) to support the claim of electron precipitation in producing storm-time NO molecules. The authors also mention that NO molecules are more numerously produced when the CME-driven storms are preceded by interplanetary shocks.

Knipp, D. J., Pette, D. V., Kilcommons, L. M., Isaacs, T. L., Cruz, A. A., Mlynczak, M. G., Hunt, L. A., & Lin, C. Y. (2017). Thermospheric nitric oxide response to shock-led storms. *Space Weather*, 15 (2), 325-342. https://doi.org/10.1002/2016SW001567

**Minor comments**

Caption of figure 2: repectively → respectively.

Caption of Table 1. "TIMMED/SEE" → TIMED/SEE.

Line 378. "author's" → authors.

Spell out DMSP the first time it is mentioned. The same for LST.